# A Novel m7G-Related Gene Signature Predicts the Prognosis of Colon Cancer

**DOI:** 10.3390/cancers14225527

**Published:** 2022-11-10

**Authors:** Jing Chen, Yi-Wen Song, Guan-Zhan Liang, Zong-Jin Zhang, Xiao-Feng Wen, Rui-Bing Li, Yong-Le Chen, Wei-Dong Pan, Xiao-Wen He, Tuo Hu, Zhen-Yu Xian

**Affiliations:** 1Department of Colorectal Surgery, The Sixth Affiliated Hospital, Sun Yat-sen University, Guangzhou 510655, China; 2Guangdong Provincial Key Laboratory of Colorectal and Pelvic Floor Diseases, The Sixth Affiliated Hospital, Sun Yat-sen University, Guangzhou 510655, China; 3Department of Radiotherapy, The Sixth Affiliated Hospital, Sun Yat-sen University, Guangzhou 510655, China; 4Department of Pancreatic Hepatobiliary Surgery, The Sixth Affiliated Hospital, Sun Yat-sen University, Guangzhou 510655, China

**Keywords:** N7-methylguanosine (m7G) methylation, m7G-related genes, colon cancer, prognostic model

## Abstract

**Simple Summary:**

N7-methylguanosine (m7G) plays an important role in the tumorigenesis and progression of colon cancer (CC). According to the capability of m7G-related genes, they are classified into three types: methyltransferases, binding proteins and demethylases. Hence, m7G-related genes could promote cancers by regulating RNAs. To further explore the functions of m7G, 29 m7G-related genes were selected and then 15 of them were utilized to construct a novel signature, termed the m7G score. Altogether, we found that the prognosis of CC patients with distinct m7G scores were significantly different. Furthermore, we applied various experiments and bioinformatics analyses to validate our results. We expect that the m7G score could indicate the correct clinical situation, which might optimize our treatments for CC patients.

**Abstract:**

Colon cancer (CC), one of the most common malignancies worldwide, lacks an effective prognostic prediction biomarker. N7-methylguanosine (m7G) methylation is a common RNA modification type and has been proven to influence tumorigenesis. However, the correlation between m7G-related genes and CC remains unclear. The gene expression levels and clinical information of CC patients were downloaded from public databases. Twenty-nine m7G-related genes were obtained from the published literature. Via unsupervised clustering based on the expression levels of m7G-related genes, CC patients were divided into three m7G clusters. Based on differentially expressed genes (DEGs) from the above three groups, CC patients were further divided into three gene clusters. The m7G score, a prognostic model, was established using principal component analysis (PCA) based on 15 prognosis-associated m7G genes. KM curve analysis demonstrated that the overall survival rate was remarkably higher in the high-m7G score group, which was much more significant in advanced CC patients as confirmed by subgroup analysis. Correlation analysis indicated that the m7G score was associated with tumor mutational burden (TMB), PD-L1 expression, immune infiltration, and drug sensitivity. The expression level of prognosis-related m7G genes was further confirmed in human CC cell lines and samples. This study established an m7G gene-based prognostic model (m7G score), which demonstrated the important roles of m7G-related genes during CC initiation and progression. The m7G score could be a practical biomarker to predict immunotherapy response and prognosis in CC patients.

## 1. Introduction

CC is one of the most common cancer types with relatively high mortality. Recently, with the rapid development of colonoscopy and CC therapy, the morbidity and mortality of CC patients have gradually decreased in some developed countries [1,2]. However, effective prognostic biomarkers are still lacking, which could improve the clinical management of CC patients. Hence, the exploration of new biomarkers with prognostic value is especially important. 

N7-methylguanosine (m7G), an important post-transcriptional modification, occurs at the N7 atom of RNA guanine by addition of a methyl group [3]. The m7G modification has been reported to promote cancer progression by modifying tRNA, miRNA and lncRNA, including the progression of intrahepatic cholangiocarcinoma, breast cancer, and lung adenocarcinoma [4,5,6]. Based on their functions in biological processes, m7G-related genes are classified into three types: writers (methyltransferases), readers (binding proteins), and erasers (demethylases) [3,7,8]. In human beings, methyltransferase-like 1 (METTL1) and the WD repeat domain 4 (WDR4) are the most well-studied regulators of m7G, and have been reported to participate in tumor progression by regulating tumor immunity, metabolic reprogramming, and drug resistance [9,10,11,12,13].

As demonstrated in the published literature, several genes are involved in the regulation of the m7G process. AGO2 has been reported to inhibit stable translation by binding to the cap located on target mRNA [14]. Additionally, translation efficiency and mRNA nuclear export are regulated by eIF4E, which could bind to the m7G cap directly [15]. Wang et al. demonstrated that DCP2 plays a vital role in decapping m7G caps on mRNA [16]. Although the above research has demonstrated the biological characteristics of m7G in detail, the prognostic value of m7G in CC patients remains elusive.

Our study constructed a m7G gene-based prognostic model (the m7G score) for CC patients based on the data from the TCGA and GEO public databases. The predictive capability of the m7G score was evaluated by KM survival curve analysis. Moreover, we have validated the expression of 15 prognosis-related m7G genes in cell lines and CC tissues from our institution. Altogether, our study identified the m7G score as a useful tool for prognosis prediction and clinical treatment guidance for CC patients.

## 2. Materials and Methods

### 2.1. Data Collection

The gene expression data, clinical characteristics, and mutational information of CC samples were obtained from The Cancer Genome Atlas (TCGA) and Gene Expression Omnibus (GEO). TCGA-COAD (*n* = 514) was downloaded from https://portal.gdc.cancer.gov up to 5 July 2022 and converted from FTPM into TPM. GSE39582 (*n* = 585) was downloaded from https://www.ncbi.nlm.nih.gov/geo/ up to 5 July 2022. Copy number variations (CNVs) of CC patients were downloaded from http://xena.ucsc.edu, accessed on 5 July 2022. The data from TCGA-COAD and GSE39582 were merged to form a new dataset via an R package for subsequent analysis. As illustrated in Appendix A, a total of 29 m7G-related genes were extracted from the published literature [5]. 

### 2.2. Unsupervised Clustering of 29 m7G-Related Genes

Among 29 m7G-related genes, there were 3 writers (METTL1, WDR4, and NSUN2), 8 erasers (DCP2, DCPS, NUDT10, NUDT11, NUDT16, NUDT3, NUDT4, and NUDT4B), and 18 readers (AGO2, CYFIP1, EIF4E, EIF4E1B, EIF4E2, EIF4E3, GEMIN5, LARP1, NCBP1, NCBP2, NCBP3, EIF3D, EIF4A1, EIF4G3, IFIT5, LSM1, NCBP2L, and SNUPN). The STRING database (http://www.db.org/ (accessed on 7 November 2022)) was utilized to analyze the interactive network of these m7G-related genes. Unsupervised clustering analysis was conducted via the Consensus Cluster Plus package (version 1.58.0) based on the k-means algorithm to evaluate the distinct expression of m7G-related genes or prognostic genes [17]. Then, CC patients were divided into three m7G clusters based on the expression level of m7G-related genes.

### 2.3. Gene Set Variation Analysis (GSVA)

Gene set variation analysis (GSVA) was performed to identify the distinct biological processes between expression signatures of m7G-related genes from different clusters via the GSVA R package (version 1.42.1). The c2.cp.kegg.V7.2.symbols gene set, downloaded from the Molecular Signatures Database, was used for GSVA. A p value less than 0.05 was considered statistically significant in the GSVA analysis [18]. 

### 2.4. Differentially Expressed Genes (DEGs)

Based on the expression level of m7G-related genes, CC patients were divided into 3 different m7G clusters. Then, DEGs among these three m7G clusters were evaluated via the R package “limma” (version 3.50.3) [19]. 

### 2.5. Gene Ontology (GO)

GO was used for enrichment analyses via the R package “cluster Profiler” (version 4.2.2). Differentially expressed genes with a *p* value < 0.05 were selected for GO enrichment pathway analysis [20].

### 2.6. Construction of Gene Clusters and m7G Score

A random forest was selected to delete redundant DEGs obtained from the previous step. Then, the prognostic significance of remaining genes was assessed via univariate cox regression analysis. CC patients were divided into 3 different gene clusters for subsequent analysis based on DEGs with prognostic significance using unsupervised clustering analysis. Then, principal component analysis (PCA) was conducted to quantify the expression signature of 15 DEGs with prognostic significance, termed the m7G score. The m7G score was established and formulated as follows:m7G score = ∑PC1i + ∑PC2i
PC1 and PC2 are principal component 1 and principal component 2 respectively, while i means the expression level of DEGs with prognostic significance among three gene clusters. The optimal cutoff was selected to divide CC patients into high- and low-m7G score groups. 

### 2.7. Single Sample Gene Set Enrichment Analysis (ssGSEA)

ssGSEA was used to evaluate the infiltration of immunocytes in CC patients among different m7G clusters, which demonstrated the correlation between m7G-related gene expression and immunotherapy response. 

### 2.8. RNA Extraction and Quantitative Polymerase Chain Reaction (qPCR)

To further validate the prognostic value of the m7G score, we assessed the expression level of 15 m7G-related genes in human CC cell lines and tissue samples. CC tissues and adjacent normal tissues were obtained from the tissue bank of Sixth Affiliated Hospital, Sun Yat-sen University. An RNA extraction process was performed using TRIzol to collect the total RNA from cell lines and tissue specimens. The reverse transcription reaction was performed using a ReverTra Ace qPCR RT Kit (Toyobo, Japan). The real-time PCR was conducted based on cDNA obtained from the above reverse transcription reaction using ABI QuantStudio™ 7 Flex Real Time PCR Systems. The expression level of m7G-related genes was normalized to β-actin using the 2^−ΔΔCt^ method. The primer sequences of the indicated genes were listed in Appendix A.

## 3. Results

### 3.1. The Alterations and Biological Characteristics of 29 m7G-Related Genes in CC Patients

A total of 29 m7G-related genes were ultimately selected based on previous studies and the expression matrices of 29 m7G-related genes were obtained from the TCGA and GEO databases (Appendix A). The regulation network diagram of these m7G-related genes is exhibited in Figure 1A. As visualized by the loop graph, the m7G-related genes were scattered across most chromosomes except for chromosomes 7, 13, 14, 18, 19, and 20 (Figure 1B). The mutation information of m7G-related genes in CC samples are displayed in Figure 1C. Overall, 102 out of 454 (22.47%) samples harbored different types of mutations, with EIF4G3 as the most common mutated m7G-related gene. Meanwhile, no mutation was identified in NUDT16, NUDT4, and NUDT4B, and missense mutations were found in EIF4E, EIF4A1, and SNUPN. Among all mutation types, missense mutation was the most frequent mutation type found in CC patients. Copy number amplification was demonstrated in 13 m7G-related genes (AGO2, LSM1, NCBP2, NSUN2, NUDT3, EIF4E1B, EIF3D, METTL1, EIF4E3, NUDT16, LARP1, GEMIN5, and NCBP3), while copy number deletion was found in 11 m7G-related genes (EIF4E2, SNUPN, IFIT5, DCPS, NCBP1, EIF4G3, EIF4E, NUDT4, DCP2, CYFIP1, and EIF4A1) (Figure 1D). As illustrated in Figure 1E, the expression levels of the indicated m7G-related genes were significantly different between CC and normal samples except for NUDT4B, CYFIP1, NCBP3, and IFIT5. Moreover, the differential expression levels of m7G-related genes in CC patients with wild EIF4G3 and mutated EIF4G3 are demonstrated in Appendix A. Taken together, these data indicate that copy number variation might be one of the regulatory mechanisms for m7G-related genes’ expression.

### 3.2. The m7G-Related Colon Cancer Subtype and Clinical Prognosis

Data from TCGA-COAD and GSE39582 were merged to form a new dataset via an R package for subsequent analysis. Based on the expression level of 29 m7G-related genes, k-means clustering was performed with different k values (k = 2–5). The best clustering effect was obtained with a k value of three (Figure 2A,B and Appendix A). CC patients were divided into three groups based on the expression level of 29 m7G-related genes, named m7G cluster A, B, and C. Moreover, based on the expression level of DEGs among the above three m7G clusters, 1535 intersecting genes were obtained and displayed in a Venn diagram (Figure 2C, Appendix A). PCA results demonstrated that the clustering result was significantly different and effective (Figure 2D). Kaplan–Meier (KM) curves demonstrated that the overall survival (OS) of CC patients in m7G clusters A and B was much better than that in m7G cluster C (Figure 2E). The heatmap illustrated m7G-related gene expression levels in CC patients with different clinical characteristics and m7G clusters (Figure 2F).

### 3.3. Biological Differences among CC Patients from Three m7G Clusters

To further identify the biological difference among CC patients in different m7G clusters, GO, GSVA, and ssGSEA analyses were performed. GO analysis results revealed that DEGs among m7G clusters were mainly enriched in the following pathways: organelle fission and nuclear division (Biological Process); chromosomal region and centromeric region (Cellular Component); and GTPase regulator activity and nucleoside-triphosphatase regulator activity (Molecular Function, Figure 3A,B). These results suggested that the DEGs among the three m7G clusters might be involved in metabolic reprogramming of CC cells. 

As shown in Figure 3C–E, the significantly dysregulated pathways between CC patients from different m7G clusters are listed. Among them, cell proliferation and metastasis-associated pathways like DNA replication, cell adhesion, and VEGF signaling were significantly enriched in m7G Cluster C. In addition, some metabolic reprogramming pathways, such as arachidonic acid metabolism, glycosphingolipid biosynthesis, and pyrimidine metabolism were remarkably upregulated in m7G Cluster C. The immune cell infiltration level was estimated by ssGSEA to further explore the difference in tumor microenvironment (TME) among CC patients from three m7G clusters. As shown in Figure 3F, infiltration of 23 immunocyte subtypes was remarkably different among CC patients from different m7G clusters.

### 3.4. Identification of Prognostic DEGs and Construction of Gene Clusters

Based on the DEGs among the three m7G clusters obtained above (Figure 2C), univariate cox analysis was performed and identified 211 DEGs with prognostic significance in CC patients (Appendix A). Based on the expression level of 211 prognostic DEGs, k-means clustering was conducted with k value ranging from two to nine. The best grouping effect was obtained with a k value of three (Figure 4A,B and Appendix A). Therefore, the CC patients were divided into three groups, termed gene clusters A, B and C. As with the survival analysis results based on the m7G clusters, KM curves indicated that the OS of CC patients in gene cluster C was much worse than those in the other two groups (Figure 4C). Moreover, the expression level of m7G-related genes was significantly different among the three gene clusters (Figure 4D). The heatmap illustrated expression levels of prognostic DEGs in CC patients with different m7G clusters and gene clusters (Figure 4E). These results validly demonstrated the remarkable effectiveness of the prognostic DEGs-based clustering.

### 3.5. Construction and Validation of the m7G Score Prognostic Risk Model

#### 3.5.1. Construction and Bioinformatic Verification of the m7G Score Prognostic Model

To further investigate the prognostic value of m7G genes, we constructed the m7G score based on the above m7G Clusters. The Sankey diagram was constructed to illustrate the modeling process (Figure 5A). Multivariate Cox regression analysis was utilized for 211 prognostic DEGs identified by univariate cox analysis in Appendix A. Furthermore, AGO2, CYFIP1, EIF4E, EIF4E2, EIF4E3, GEMIN5, METTL1, NCBP1, NSUN2, NUDT10, NUDT11, NUDT3, NUDT4, SNUPN, and WDR4 (Appendix A) were chosen for m7G prognostic model construction by the formula: m7G score = ∑PC1i + ∑PC2i. 

As demonstrated in Figure 5B,C, the m7G score was remarkably different in CC patients among different m7G clusters and gene clusters, which indicated the significant effectiveness of m7G score construction. Then, CC patients were divided into high- and low-m7G score groups by an optimal cutoff. The OS of CC patients in the high-m7G score group was much better than that of the low group (Figure 5D). Furthermore, subgroup analysis results revealed that the survival rate difference was much more significant in CC patients with T3–T4 cancer, which illustrated that the prognostic prediction power of the m7G score was stronger in advanced CC patients (Figure 5E,F). 

More importantly, the performance of the m7G score was further validated in another external GEO cohort (GSE31595). CC patients were divided into high-m7G score and low-m7G score groups utilizing the same grouping method. As shown in Figure 5G, the survival rate of CC patients with a high m7G score was much higher than that of low-m7G score patients. Taken together, the m7G score is a robust prognostic model with excellent predictive power for CC patients.

#### 3.5.2. Validation in Human CC Cell Lines by qPCR Assay

We further evaluated the expression level of 15 m7G prognostic genes used for m7G score construction in human CC cell lines and tissues using a qPCR assay. As demonstrated in Figure 6A–O, the mRNA levels of AGO2, CYFIP1, EIF4E, METTL1, NSUN2, NUDT3, SNUPN, NUDT4, GEMIN5, EIF4E2, NCBP1, and WDR4 were remarkably higher in CC cell lines (HCT15, DLD1, RKO, HCT8, HCT116, SW48, and WiDr) than those in the normal cell line HIEC6. By contrast, the expression levels of EIF4E3, NUDT10 and NUDT11 were significantly lower in CC cell lines. 

#### 3.5.3. Validation in Human CC Tissues by qPCR Assay

More importantly, the mRNA levels of 15 m7G prognostic genes were assessed in 15 matched CC and normal tissues from our tissue bank. Similar expression patterns for these m7G prognostic genes were obtained in human CC tissues by qPCR results (Figure 7A–O). These results further validated the robust efficiency of the m7G score.

### 3.6. Drug Sensitivity in High- and Low-m7G Score Groups

To further explore differences in drug resistance in CC patients with high and low m7G scores, we assessed the estimated IC50 levels of chemotherapy drugs or inhibitors in the above two groups. As demonstrated in Figure 8, CC patients with a high m7G score were found to be more sensitive to Vinblastine, BIBW2992, Cytarabine, Docetaxel, Erlotinib, Paclitaxel and Rapamycin, while patients with a low m7G score responded better to AP.24534, Bleomycin, Cisplatin, Doxorubicin, Embelin, Gefitinib, Meformin and Pazopanib. Altogether, these data revealed that the m7G score could also be a potential indicator for drug sensitivity in CC patients.

### 3.7. Tumor Microenvironment in High- and Low-m7G Score Groups

Since the m7G score was associated with the prognosis of CC patients, the correlation between m7G score and tumor microenvironment was further assessed. The tumor mutation burden (TMB) was found to be negatively correlated with m7G score in CC patients (R = −0.13, *p* = 0.0085; Figure 9A). More importantly, the PD-L1 expression level of CC patients in the low-m7G score group was significantly higher than that in the high group, which demonstrated that the m7G score could serve as an indicator for predicting anti-PD1/PD-L1 immunotherapy response (Figure 9B).

The CIBERSORT algorithm was applied to further identify the correlation between immune infiltration and m7G score. Negative correlations between m7G score and immune infiltration were identified in several immunocytes, including B cells, naive B cells, Macrophages, M1 Macrophages, M2 Macrophages, Myeloid dendritic cells, Neutrophils, CD4+ T cells and CD4+ memory resting T cells. Meanwhile, positive correlations between m7G score and immune infiltration were observed in plasma B cells, CD8+ T cells and regulatory T cells (Figure 9C–N). Therefore, the m7G score was significantly associated with immune cell infiltration in CC patients.

## 4. Discussion

Colon cancer (CC), one of the common malignant tumors worldwide, is a serious challenge to the safeguarding of human health. The current prognostic index is still insufficient to evaluate the prognosis of CC patients in clinical work. Although some research has demonstrated the potential role of m7G genes in the tumorigenesis of several tumors, including acute myeloid leukemia, bladder cancer, and esophageal squamous cell carcinoma, the prognostic value of m7G-related genes is still unclear in CC patients [1,2,21]. Therefore, constructing an effective prognostic model based on m7G genes is of great clinical importance. In this study, the m7G score prognostic model was established via unsupervised clustering and PCA analysis based on data from TCGA-COAD and GSE39582. The GSVA, ssGSEA, GO, KEGG, and KM curve analyses were utilized to identify the biological characteristics of CC patients with different m7G scores. Moreover, the expression levels of 15 prognosis-related m7G genes was further confirmed in human CC cell lines and tissues by qPCR.

Several previous studies have demonstrated the predictive value of m7G-related long noncoding RNAs (lncRNAs) and microRNAs (miRNAs) in different types of cancer. Hong and Du et al. showed that m7G-related miRNA was related to cancer cell migration, tumor immunity, and prognosis [22,23]. Additionally, the predictive value of m7G-related lncRNA has been reported by Ming and Wang, which could be used to assess oncogenesis and treatment response in renal clear cell carcinoma and bladder cancer [24,25,26]. These studies have inspired us to explore whether m7G-related genes have a similar prognostic prediction effect. The present study constructed the m7G score on the basis of 15 prognostic m7G genes (AGO2, CYFIP1, EIF4E, EIF4E2, EIF4E3, GEMIN5, METTL1, NCBP1, NSUN2, NUDT10, NUDT11, NUDT3, NUDT4, SNUPN, and WDR4) [14,27,28,29,30]. In detail, CC patients with higher m7G scores obtained lower PD-L1 expression levels as well as better prognoses.

Tumor mutational burden (TMB) has been reported as an indicator for immunotherapy efficacy in cancer patients, including CC, melanomas, renal cell carcinomas, bladder cancers as well as head and neck squamous cell cancers [27,28,29,30,31,32,33]. Usually, cancer patients with high TMB respond better to immunotherapy than those with low TMB [27,28,29,30,31,32]. Nevertheless, the role of TMB in CC patients remains controversial. Liu et al. illustrated that CC patients with high TMB exhibited a higher OS rate than those with low TMB [27]. In contrast, Zhou et al. demonstrated that CC patients with low TMB had better prognoses than their counterparts in the high TMB group [33]. In accordance with Zhou’s findings, our data indicated that CC patients with higher m7G scores obtained lower TMB and higher OS rates. Hence, the correlation between TMB and m7G is worthy of further investigations.

PD-1, located on membrane of T cells, is a well-known immune checkpoint and participates in the immune escape of cancer cells by binding to PD-L1 on the tumor cell surface. Hence, antitumor immunotherapy was developed based on PD-1/PD-L1 blockages or inhibitors, which has achieved tremendous success in cancer patients [34]. Generally, patients with rich but exhausted immunocyte infiltration respond better to PD-1/PD-L1 inhibitors than their counterparts [35,36,37]. In this study, CC patients in the high-m7G score group exhibited lower expression levels of PD-L1, indicating that m7G-related genes might regulate PD-1/PD-L1 expression and thereby affect the response to anti-PD-1/PD-L1 therapy. 

There are some limitations to the present study. To begin with, most data involved in our study were downloaded from public databases, which may inevitably lead to uncertain selection bias. Although we have primarily proven the prognostic value of the m7G score in an external validation cohort and demonstrated differential expression levels of 15 prognostic m7G genes in CC patient samples from our institution, further validation work based on CC cohorts from multiple centers is needed. 

## 5. Conclusions

In summary, our study constructed an m7G score prognostic model for CC patients. The m7G score could play an important role in prognosis prediction and immunotherapy evaluation, which could offer significant benefits in the clinical management of CC patients.

## Figures and Tables

**Figure 1 cancers-14-05527-f001:**
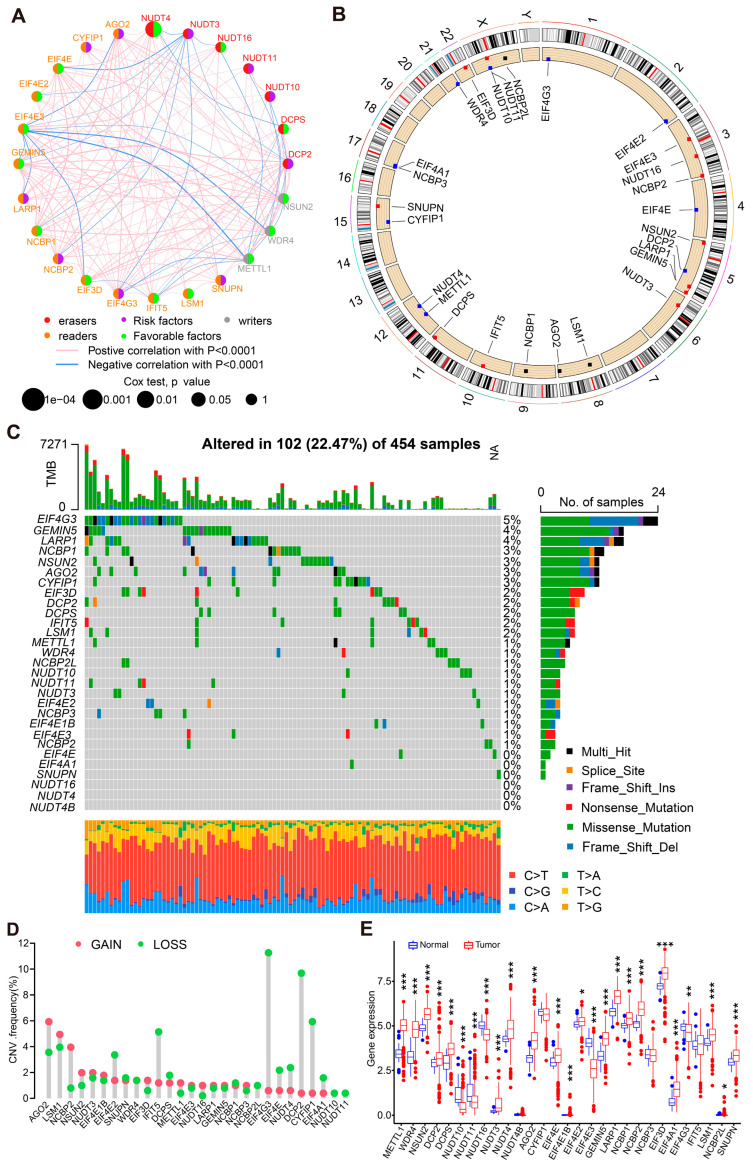
Genetic alterations and expression of the 29 m7G-related genes in CC patients. (**A**) Expression modification of m7G-related genes and their effect on regulation. (**B**) Gene location on chromosome with mutation information. Blue dots indicate deletion and red dots mean amplification. (**C**) Copy number variation (CNV) of m7G-related genes in CC samples; the mutation frequency is listed on the right. (**D**) Copy number of each m7G-related gene in detail. GAIN refers to copy number amplification and LOSS means copy number deletion. (**E**) The boxplot for the differentially expressed m7G-related genes between normal and CC samples. * *p* < 0.05, ** *p* < 0.01, *** *p* < 0.001.

**Figure 2 cancers-14-05527-f002:**
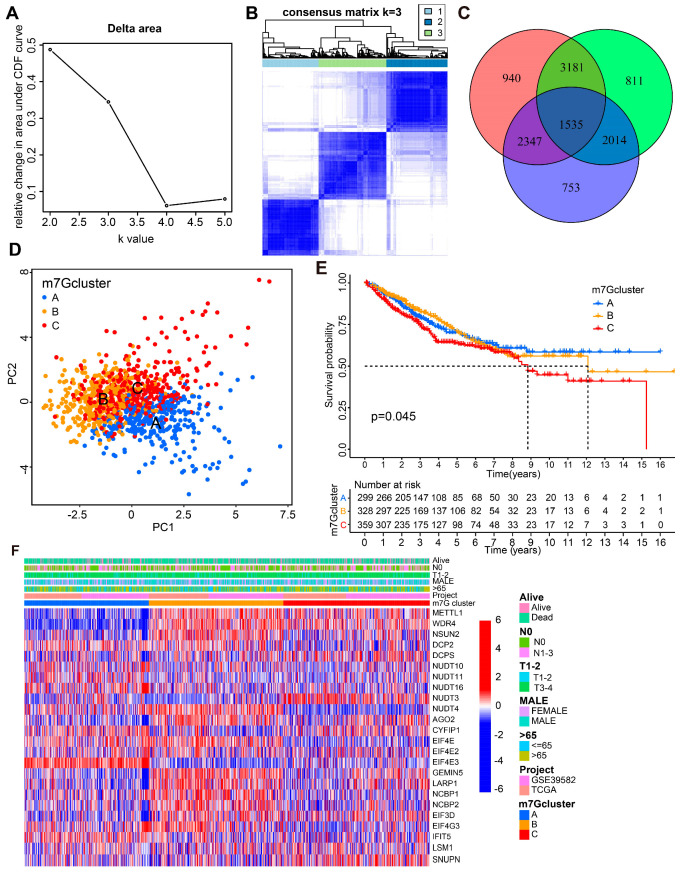
The construction of m7G-related subtype clusters. (**A**) Cumulative distribution function curve illustrates the most effective way of m7G clustering. (**B**) The consensus matrix of the clustering analysis based on m7G expression profiles via k-means clustering (k = 3). (**C**) The Venn diagram depicted the intersection of differentially expressed genes among different m7G clusters. (**D**) The principal component analysis (PCA) for m7G clusters. (**E**) Kaplan–Meier (KM) curves for the overall survival (OS) of CC patients among different m7G groups. (**F**) Heatmap of m7G-associated genes’ expression in CC patients with different clinical characteristics, data sources and m7G clusters.

**Figure 3 cancers-14-05527-f003:**
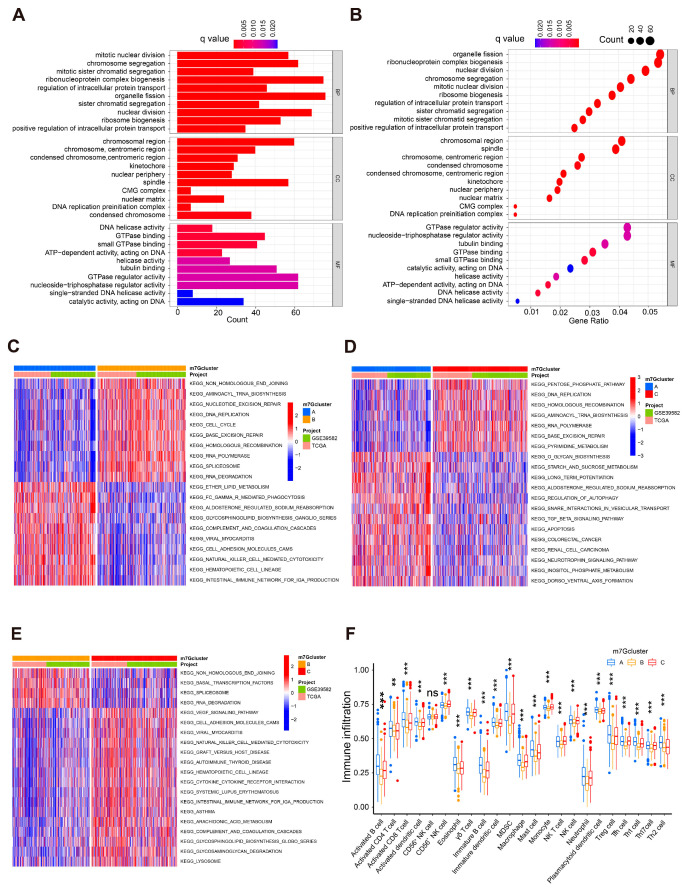
Biological characteristics and immunocyte infiltration information of three m7G Clusters. (**A**,**B**) GO analysis for the differentially expressed genes from different m7G clusters. (**C**–**E**) Heatmaps of the remarkably different pathways among different m7G groups by GSVA analysis. (**F**) The boxplot for immune infiltration among CC patients from different m7G groups. * *p* < 0.05, ** *p* < 0.01, *** *p* < 0.001.

**Figure 4 cancers-14-05527-f004:**
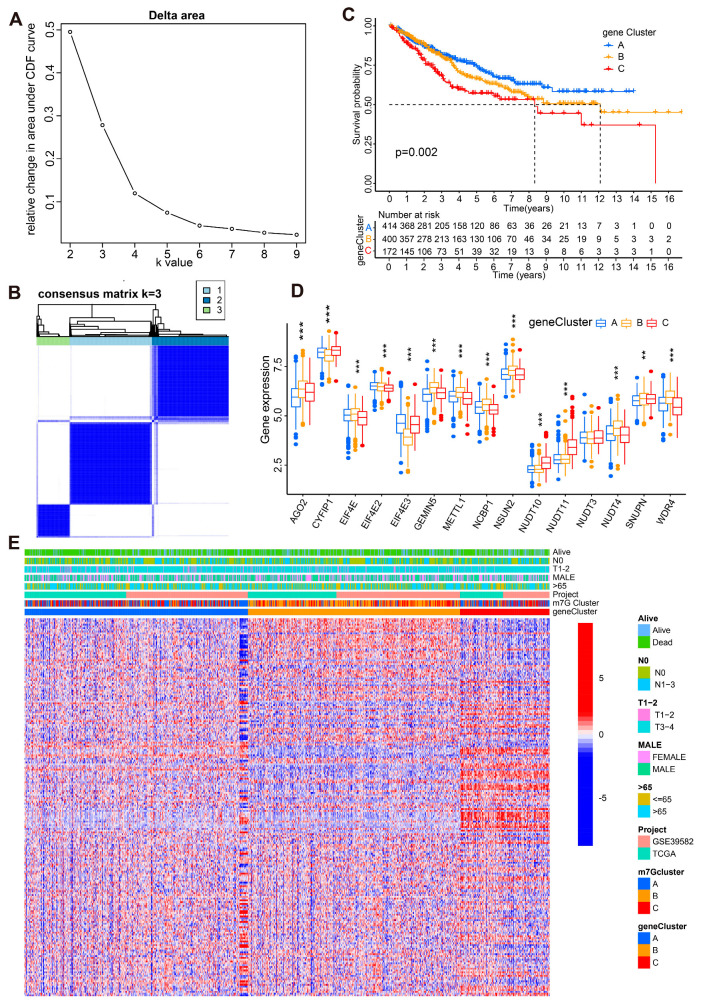
Gene clustering based on prognostic m7G-related DEGs in CC patients. (**A**) Cumulative distribution function curve demonstrates the most effective way of gene clustering. (**B**) The consensus matrix of the clustering analysis based on prognostic m7G-related gene expression profiles via k-means clustering (k = 3). (**C**) KM curves of OS among CC patients from different gene clusters. (**D**) The boxplot for m7G-related genes’ expression levels among CC patients from different gene clusters. (**E**) Heatmap depicting expression levels of prognostic m7G-related genes in CC patients with different clinical characteristics, data sources, m7G clusters, and gene clusters. * *p* < 0.05, ** *p* < 0.01, *** *p* < 0.001.

**Figure 5 cancers-14-05527-f005:**
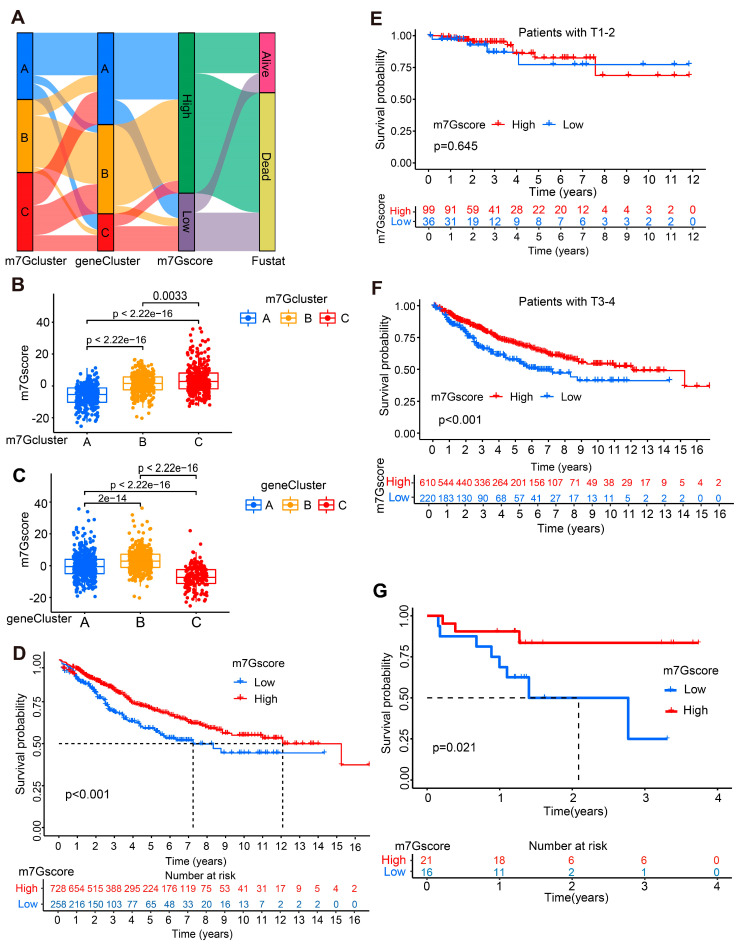
Construction and verification of the m7G score prognostic model. (**A**) Sankey diagram for the modeling process. (**B**) The m7G score level in CC patients from different m7G clusters. (**C**) The m7G score level in CC patients from different gene clusters. (**D**) KM curves for the OS of CC patients from high- and low-m7G score groups. (**E**,**F**) KM survival analysis based on m7G score in CC patients at different T stages. (**G**) KM survival curves based on m7G scores in CC patients from the GSE31595 validated cohort.

**Figure 6 cancers-14-05527-f006:**
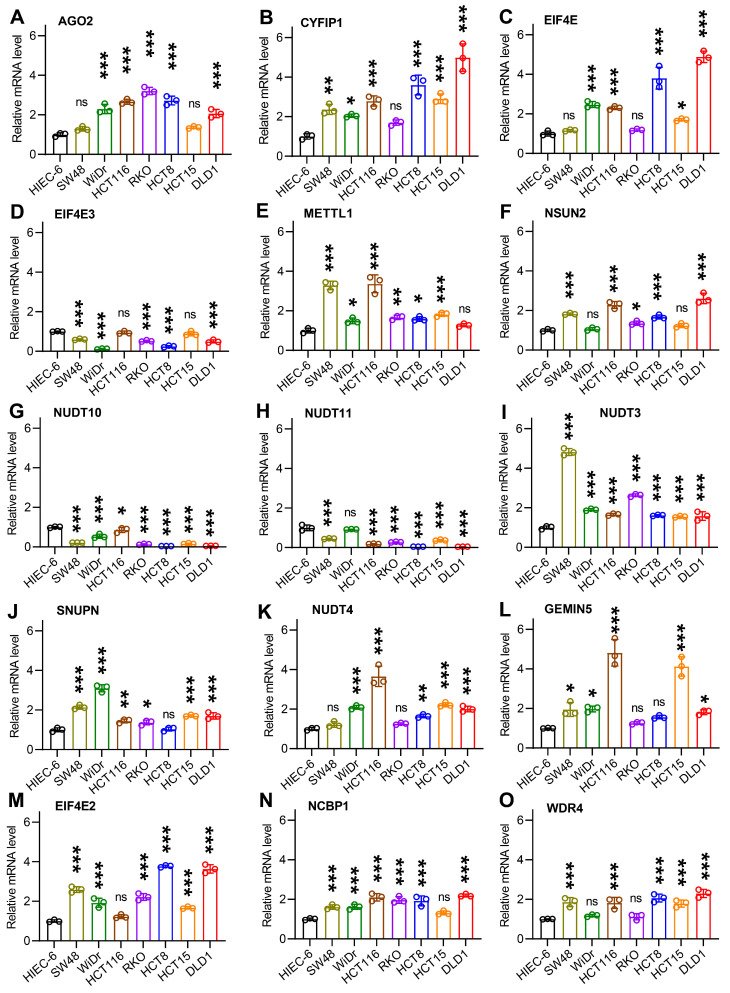
Validation for the expression level of 15 prognostic m7G-related genes in human CC cell lines. (**A**–**O**) Expression level of 15 prognosis-associated m7G genes in 7 human CC cell lines (RKO, HCT8, HCT116, SW48, WiDr, HCT15, and DLD1) and normal human intestinal epithelial cells (HIEC6). * *p* < 0.05, ** *p* < 0.01, *** *p* < 0.001.

**Figure 7 cancers-14-05527-f007:**
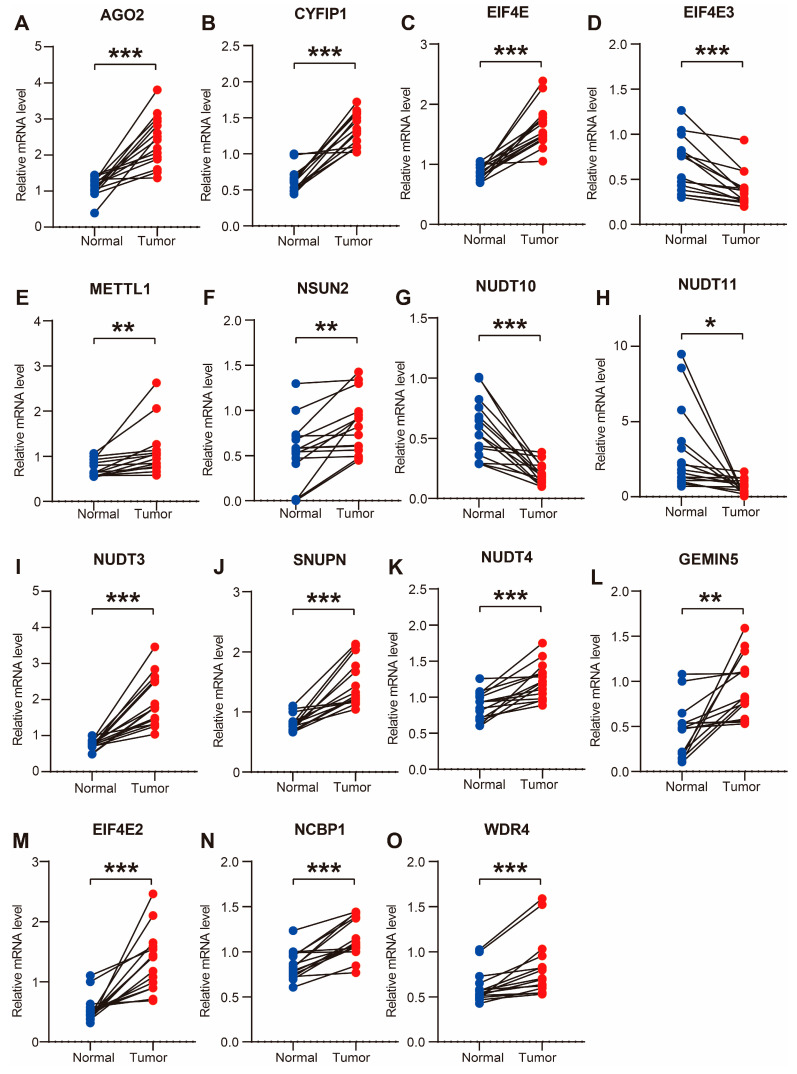
Validation of the expression level 15 prognostic m7G-related genes in human CC tissues. (**A**–**O**) Expression level of prognostic m7G-related genes in 15 paired human CC and adjacent normal tissues. * *p* < 0.05, ** *p* < 0.01, *** *p* < 0.001.

**Figure 8 cancers-14-05527-f008:**
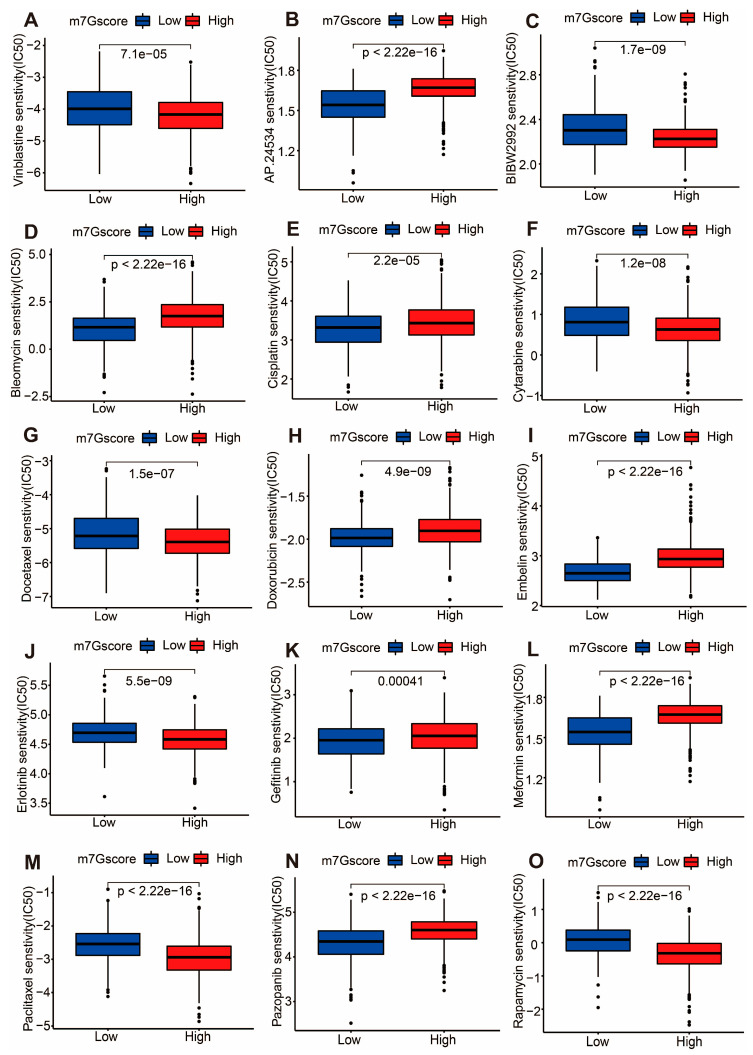
Drug sensitivity in CC patients from different m7G score groups. (**A**–**O**) Boxplots depicting the IC50 value of Vinblastine (**A**), AP.24534 (**B**), BIBW2992 (**C**), Bleomycin (**D**), Cisplatin (**E**), Cytarabine (**F**), Docetaxel (**G**), Doxorubicin (**H**), Embelin (**I**), Erlotinib (**J**), Gefitinib (**K**), Meformin (**L**), Paclitaxel (**M**), Pazopanib (**N**), and Rapamycin (**O**) in CC patients with different m7G scores.

**Figure 9 cancers-14-05527-f009:**
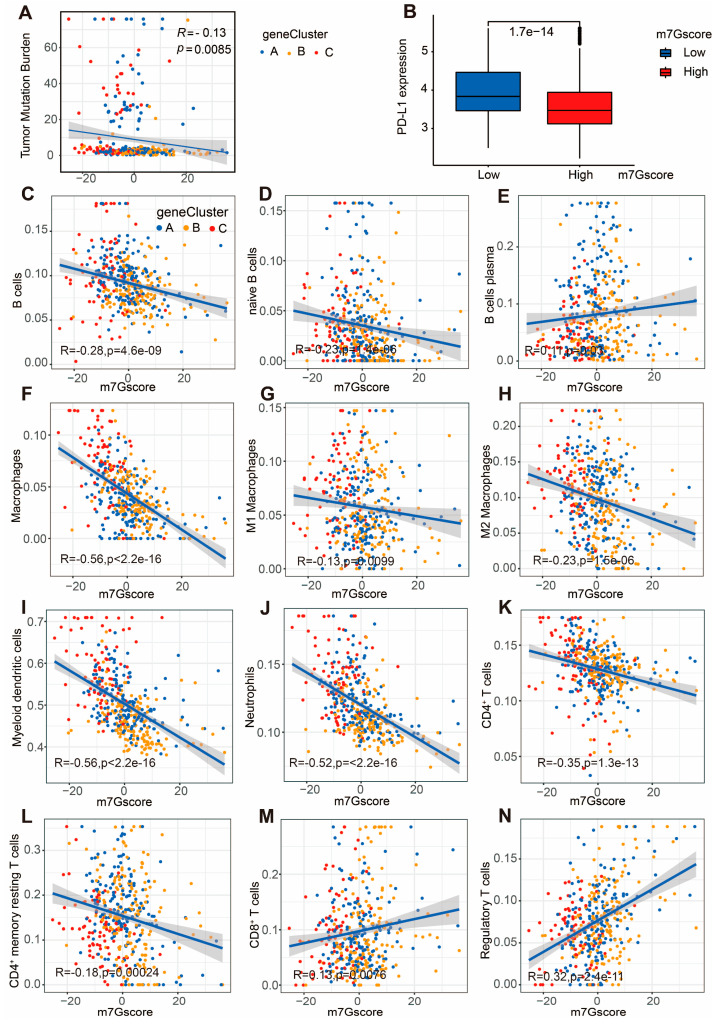
Correlation analysis between m7G score and TMB, as well as tumor immune microenvironment. (**A**) Relevance of TMB and m7G score in CC patients. (**B**) Expression level of immune checkpoint PD-L1 in CC patients from high- and low-m7G score groups. (**C**–**N**) Correlation analysis between infiltration levels of B cells (**C**), naive B cells (**D**), B cells plasma (**E**), Macrophages (**F**), M1 Macrophages (**G**), M2 Macrophages (**H**), Myeloid dendritic cells (**I**), Neutrophils (**J**), CD4+ T cells (**K**), CD4+ memory resting T cells (**L**), CD8+ T cells (**M**), regulatory T cells (**N**) and m7G score.

## Data Availability

The processed data that support the findings of this study are available from the corresponding author.

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
