# Peer review of "A Novel m7G-Related Gene Signature Predicts the Prognosis of Colon Cancer"

_cancers, 2022, doi:10.3390/cancers14225527_

Round 1
Reviewer 1 Report
This manuscript by Chen et al examines the role of N7-methylguanosine (m7G) gene as a diagnostic and prognostic biomarker in colon cancer (CC). The authors have critically examined the publically available database for m7G and corroborated their data with a set of the clinical sample. the manuscript is well written and I do not see any error in the writeup.
Author Response
We truly appreciate the reviewer’s positive and constructive comments on our work. Thank you again for your support.
Reviewer 2 Report
In this present study, the authors have very methodically analyzed a prognostic marker for colon cancer in the form of methylation status of the RNA guanine. They have very carefully analyzed the clinical patients data from publicly available databases. Among the 29 m7G genes, they have analyzed and identified 15 as having prognostic potential. They have further validated the expression level of these genes in multiple human cell lines and 15 patient samples (with adjacent controls). They generated m7G scoring patterns and correlated with the high survival rate with increasing score. They also correlated their data with drug sensitivity. Thus, these data could be further tested for a reliable prognostic marker, the need of the hour.
This reviewer finds the study very timely, appropriate, well designed, supported by data and statistical significance tests.
This reviewer have a couple of clarifications/suggestions that can be addressed by the authors.
1. In the patient sample analysis, the authors had no comment on the gender of the patients. Did they consider gender? How many of each gender used?
2. They used the adjacent normal tissue as their control for qPCR analysis. Did they consider age matched colon tissues from post-mortem patients without CC diagnosis as a control (if available)?
3. Did they check a few of the 14 m7G genes that did not show prognostic potential to check the non-correlation at least in a few human cell lines?
They may not need to perform experiments, however, if they have data, they can share or at least, comment on these.
Minor mistakes:
They have repeated the information of "qPCR analysis in human CC cell lines and samples" twice in the abstract.
